# OpenReview forum: "On Almost Surely Safe Alignment of Large Language Models at Inference Time"
_TMLR — Decision pending for TMLR_

### Review · Reviewer_iMvo · 2025-12-07

**Summary Of Contributions:**

This paper introduces InferenceGuard, an inference-time decoding framework designed to produce almost surely safe LLM outputs relative to a specified safety cost model. The main contributions are:

1.	Constrained-MDP formulation of safe decoding using a safety-budget tracker and infinite penalties to encode hard safety constraints.
2.	Latent-space MDP abstraction with theoretical results establishing equivalence to the token-space augmented MDP and showing that any finite-cost optimal policy satisfies the safety constraint almost surely.
3.	A practical blockwise decoding algorithm, which combines a latent critic (predicting both safety and task reward) with block/beam expansion to prune unsafe trajectories during generation.
4.	Extensive empirical evaluation across several LLaMA-family models and standard safety datasets, demonstrating improved reward–safety trade-offs relative to BoN, beam search, ARGS, and RE-Control.

Strengths

•	Strong and elegant theoretical framing grounded in safe RL.

•	Clear articulation of the idealized almost-sure safety guarantee.

•	Practical, model-agnostic inference-time method requiring no finetuning.

•	Empirical evaluation is broad and compares against relevant baselines.

Weaknesses

•	The theoretical guarantees rely on idealized assumptions; the implemented algorithm substantially approximates them.

•	Safety performance critically depends on the correctness and coverage of the underlying safety cost model.

•	Latent-space Markov/sufficiency assumptions are not well justified for real LLMs.

**Additional Comments:**

The latent transition is deterministic given a token, yet the paper phrases it as a stochastic kernel. This creates unnecessary complexity in the Bellman optimality section and in the associated measurability assumptions. Explicitly noting that the latent transition is a Dirac delta would improve clarity.

**Audience:**

Yes

**Audience Explanation:**

•	Inference-time alignment is a rapidly growing area of interest for both ML and AI safety communities.

•	The constrained-MDP perspective is novel and of methodological interest to researchers in RL, decoding, and model steering.

•	Latent-space critics for decoding control are relevant for controllable text generation and for inference-time safety systems.

•	The empirical findings address reward–safety–latency trade-offs that matter to practitioners deploying LLMs in safety-sensitive contexts.

**Broader Impact Concerns:**

•	Safety guarantees hold only relative to the specific cost model, which may be incomplete or biased.

•	Absence of adversarial/jailbreak evaluation limits real-world robustness claims.

•	The method may encourage overconfidence in safety for high-stakes deployments.

•	Inference-time filtering may mask underlying model vulnerabilities rather than address the root causes.

**Claims And Evidence:**

Yes

**Claims Explanation:**

Theoretical support

•	The constrained-MDP formulation, augmented-state construction, and latent-MDP equivalence are correct under the stated assumptions.

•	The “almost surely safe” guarantee holds in the idealized infinite-penalty limit with optimal policies.

Empirical support

•	Experiments include several LLaMA-based models and commonly used safety datasets.

•	Comparisons to strong baselines (ARGS, RE-Control, BoN, beam search) are appropriate.

•	Reported reward, safety cost, safety rate, and latency metrics generally support the claimed improvements.

Limitations

•	There is a notable gap between the theoretical formulation and the practical decoding algorithm, which uses approximate critics, finite penalties, and heuristic beam filtering.

•	The method’s safety depends entirely on the accuracy of the cost model, yet robustness to misspecification is not analyzed.

•	The claim that (h_t, o_t) are “ideal” critic inputs is overstated; current citations do not establish sufficiency or low dimensionality in practice.

Overall:
Evidence supports the relative performance improvements and conceptual contributions, but not the stronger implication that the implemented method achieves practical “almost sure safety.”

**Requested Changes:**

Critical

1.	Clarify the theory–practice gap.
Clearly distinguish which aspects of the almost-sure safety guarantee apply to the implemented algorithm, given approximate critics, finite penalties, and finite beam sizes.

2.	Discuss dependence on the safety cost model.
Add analysis or explicit acknowledgment of how cost-model misspecification, calibration errors, or distribution shift affect safety outcomes.

Recommended

3.	Explain latent-MDP assumptions.
Briefly justify or qualify the assumption that hidden states/logits form a sufficient Markov state for safety prediction.

4.	Add practical guidance on hyperparameters and computational cost.
Help readers understand how block size d, beam size K, number of samples N, and penalty values affect performance and latency.

5.	Qualify claims about latent-space critics and parameter-free arguments.
While latent-space critics may reduce input dimensionality, the paper does not present empirical evidence of increased efficiency.
Additionally, although the augmented MDP avoids explicit Lagrange multipliers, the implemented method introduces hyperparameters (finite penalties, safety thresholds) that implicitly mediate the reward–safety tradeoff.

---

> ### Author Response · Authors · 2026-03-24
>
> We are grateful for the reviewer's deep engagement with both our theoretical framework and empirical results.
>
> > **Clarify the theory–practice gap**
>
> We would like to clarify that our theoretical results do not claim that the optimal policy with an almost sure safety guarantee is always found by our proposed InferenceGuard algorithm. Rather, the theory characterizes the *existence and properties of optimal policies* in the augmented MDP from Eqn. 7. In particular, Theorem 1 establishes the existence of optimal policies in both latent and original token spaces for all penalty values $n > 0$, and as $n \to \infty$, this implies the existence of an optimal policy that is almost surely safe with respect to the given cost model. As stated in the paper, our guarantees apply *upon solving the augmented MDP with sufficiently large penalties*, and are therefore statements about the solution of the optimization problem, not the optimization procedure itself. Moreover, solving this augmented MDP would require computing the corresponding optimal value function, with effectively infinite penalties for safety violations, and optimizing over the full policy space (see Eq. 8, Theorem-1, Prop-I). In practice, InferenceGuard only approximates this through finite penalties, finite beam widths, and learned critics that only approximately estimate future costs/rewards from available training samples. These approximations limit the extent to which the optimal solution can be recovered in practice. We note that this distinction between theoretical guarantees on the optimal solution and approximate solution methods is standard in RL and RLHF, where optimal policies or value functions are characterized theoretically but only approximated in practice. **We clarified this distinction explicitly in the new revision end of Section 5.2.**
>
> > **Discuss dependence on the safety cost model.**
>
> Our safety guarantee is explicitly defined with respect to the chosen safety cost model, as stated in the paper. Therefore, cost-model misspecification, miscalibration, or distribution shift can weaken the resulting practical safety level. In this sense, InferenceGuard does not remove the need for a good safety model; rather, it provides a principled way to enforce safety relative to that model during decoding. **We have clarified this point in the introduction of the revised paper. Empirically, we also additionally report results with two different cost models in Appendix E.4 (Table 12)**. The resulting safety-rate improvements are consistent across both cost models, and a judge-model comparison on the same task and base model yields similar helpfulness and harmlessness win rates. This suggests that the method is not tied to a single safety model and remains effective across a reasonable range of cost models. However, we do not claim robustness to arbitrarily poor or severely misspecified cost models. A more systematic study of cost-model quality, calibration, and distribution shift is an important direction for independent future work.
>
> > **Explain latent-MDP assumptions.**
>
> We refer the reviewer to Section 3, where we formalize the LLM dynamics. In particular, the generation process evolves as $[h_{t+1}, o_{t+1}] = f_{\text{LLM}}(h_t, y_t)$ with $y_t \sim \text{SoftMax}(W o_t)$, where $h_t$ aggregates the prefix through accumulated key-value states and $o_t$ parameterizes the next-token distribution. Accordingly, conditioned on $(h_t, o_t)$, the distribution over the next token is fully determined by the model, i.e., $P_\theta(y_{t+1} \mid y_{1:t}) = P_\theta(y_{t+1} \mid h_t, o_t)$. Thus, the generation process is Markov in the latent space, and we use $(h_t, o_t)$ as the state representation in our MDP formulation. Since the hidden state determines the model’s future token distribution, any property that is a function of the generated token sequence (including safety) is, in principle, determined by this latent state. This provides a natural and standard representation for predicting future costs in our framework.
>
> > **Add practical guidance on hyperparameters and computational cost.**
>
> **We added practical hyperparameter guidance in Appendix E.1.** The appendix includes the default decoding settings (Table 1), an ablation over the number of samples $N$, block size $d$, and beam size $K$ for InferenceGuard (Table 7), and an inference-time decomposition into generation and evaluation cost (Table 13). Based on these results and our empirical findings, we revised the appendix text to make these trends explicit and to provide recommended default settings along with the experimental setup: increasing the number of samples $N$ generally improves safety at the cost of higher latency, larger $d$ tends to strengthen safety while reducing reward, and more aggressive pruning via smaller $K$ reduces compute with only modest quality changes.
>
>
> (to be continued...)

---

> ### Author Response · Authors · 2026-03-24
>
> > **Absence of adversarial/jailbreak evaluation**
>
> **We added experiments on three adversarial benchmarks, *AdvBench*, *MaliciousInstruct* and *Harmbench*, using two base models, Alpaca-7B and Vicuna-7B in additional experimental results in Appendix E.3.** The results show that our method maintains strong safety under adversarial prompting, substantially improves over the base models, and is generally competitive with or better than the decoding baselines in both safety rate and ASR (Attack Success Rate).

---

### Review · Reviewer_DTpC · 2026-02-25

**Summary Of Contributions:**

The work proposes an inference-time adaptation to LLM inference -- InferenceGuard -- that limits unsafe outputs. The work relies on modelling the generation as different variants of MDPs. It then leverages beam search and a critic to penalise unsafe generation directions.

Strengths: interesting formalisation and effectiveness in the evaluated setting.

Weaknesses: limited experiment setup (models, data), lack of adversarial testing, unclear positioning.

**Audience:**

Yes

**Audience Explanation:**

Preventing generation of unsafe outputs is a critical area of LLM research.

**Claims And Evidence:**

No

**Claims Explanation:**

The core evaluation is good. However, it remains unclear how the work fits into modern inference systems. Additionally, lack of any kind of evaluation of adversarial settings is the weakest point of this work. See requested changes.

**Requested Changes:**

1. The work could use more discussion regarding how the work fits into modern inference pipelines. It isn't immediately obvious to me how this would work with batched inference with speculate decoding using a larger model. Which makes the positioning unclear. This also isn't a replacement to RLHF (for all its flaws) which is ammortised when it shifts the compute to inference time.
2. Any reason for not comparing with output judges?
3. Similarly, it seems to me that the work comes down to "good critic -> good system". It could use more elaboration.
4. I'm not sure I'm convinced by "unsafe tokens" -> harmful response. This doesn't hold universally and could use more elaboration. Related to my next point.
5. The evaluation is incomplete without adversarial prompts. All modern LLMs are quite good at avoiding harmful generation in easy settings. Especially, when you consider an adaptive setting that account for the critic / InferenceGuard.

---

> ### Author Response · Authors · 2026-03-24
>
> We are grateful for the reviewer's thorough evaluation and recognition of our formalization's strengths, and we appreciate the constructive feedback regarding the experimental scope and adversarial robustness.
>
> > **Fit into Modern Pipelines**
>
> We would like to clarify that our method operates at the decoding level and is compatible with standard inference pipelines.  In particular, InferenceGuard builds upon beam search–style decoding by augmenting the search with safety-aware state tracking and critic-based scoring, without modifying the underlying model parameters. As such, it can be integrated with batched inference by applying the same augmentation and scoring independently across batch elements if the memory capacity of the compute permits.
>
> Regarding speculative decoding, our method is orthogonal: it can be applied after the verification step, operating on candidate sequences that are consistent with the target model.
> For SOTA speculative decoding methods that build draft trees, e.g. EAGLE [1], it is already simulating beam search and therefore only the verification step has to be changed to integrate the critic scores with the large model logits.
>
> [1] Li, Yuhui et al. “EAGLE-2: Faster Inference of Language Models with Dynamic Draft Trees.” Conference on Empirical Methods in Natural Language Processing (2024).
>
> > **Replacement to RLHF**
>
> We would like to clarify that our approach is not intended as a replacement for RLHF or other training-time alignment methods. Rather, the two serve different usage scenarios and involve different trade-offs. Training-time alignment improves average-case safety by incorporating alignment into the model parameters, whereas InferenceGuard is designed for inference-time alignment with formal almost-sure safety guarantees (with respect to the cost model) for each generated response. This distinction is especially important in settings where strong safety assurances are required. A direct efficiency comparison is also inherently subtle, because the computational costs arise in different places: training-time alignment incurs a substantial one-time optimization cost, while inference-time alignment incurs additional cost at query time. If the objective is not formal safety guarantees but only improving average safety rates, then the preferred approach depends on the usage regime. For example, in settings where the training set is much larger than the number of deployment queries, or when rapid deployment without retraining is needed, inference-time alignment can be more cost-effective overall. In contrast, if a relatively small training set is sufficient and the aligned model is used for a very large number of queries, then training-time alignment may become more efficient in the long run. **We clarified this distinction more explicitly in the related works (Section 2).**
>
> > **Any reason for not comparing with output judges?**
>
> We kindly refer the reviewer to Figure 5 and Table 11 in Appendix E.4 where we evaluate the safety of the generations using a stronger model (Deepseek-R1-Distill-Qwen-32B) as a judge. We evaluated 'Helpfulness' win-rate and 'Harmlessness' win-rate.
> Although we only optimized safety through the [beaver-7b-unified-cost](https://huggingface.co/PKU-Alignment/beaver-7b-unified-cost) cost model, we achieve better helpfulness and harmlessness win rate compared to all the other baselines with respect to Deepseek-R1-Distill-Qwen-32B.
>
> > **The work comes down to "good critic -> good system".**
>
> Our work elaborates on several component, the first question being how to train a good critic given a cost model and a reward model.
> We show that this critic can be trained on top of the hidden representation of the LLM with a loss that is theoretically justified from safe reinforcement learning with augmented state.
> Once we have a good critic, how to properly search in the token space is another question we address to ensure that we will produce safe answers, as sampling only from the LLM and waiting for a safe answer is too costly.
> Therefore, we propose a way to bias and steer the LLM distribution with the trained critic in InferenceGuard.
>
> > **I'm not sure I'm convinced by "unsafe tokens" -> harmful response.**
>
> The cost model we rely on is also a textual transformers, it does not simply detect independent unsafe tokens, the safety score is attributed based on the combination and order of every token composing the sentence. You can refer to [2] for the full training procedure of this model.
>
> [2] Dai J, Pan X, Sun R, Ji J, Xu X, Liu M, Wang Y, Yang Y. Safe rlhf: Safe reinforcement learning from human feedback. ICLR 2024.
>
> (to be continued...)

---

> > ### Author Response · Authors · 2026-03-24
> >
> > > **The evaluation is incomplete without adversarial prompts.**
> >
> > **We added new experiments on three adversarial benchmarks, *AdvBench*, *MaliciousInstruct* and *Harmbench*, using two base models, Alpaca-7B and Vicuna-7B in additional experimental results in Appendix E.3.** The results show that our method maintains strong safety under adversarial prompting, substantially improves over the base models, and is generally competitive with or better than the decoding baselines in both safety rate and ASR (Attack Success Rate).
> >
> > We emphasize, however, that safety rate and ASR capture related but different notions of safety, safety rate is computed using the given cost model, so higher is better, whereas ASR is the standard jailbreak metric based on keyword or string matching used by these benchmarks, so lower is better. Our method is not designed to optimize jailbreak success directly; rather, it maximizes task quality subject to a safety constraint, and lower ASR is a desirable consequence of improved safe decoding. In practice, we also observed that some safe indirect refusal responses (e.g., "It is not advisable to do that" or "This is dangerous behavior") may still be counted as successful attacks under certain benchmark rules. Therefore, ASR is rather evaluating the refusal behavior of the base and cost models rather than the actual safety of the response. Finally, we note that PKU-SafeRLHF and HEx-PHI already contain harmful and partially adversarial prompts, although they are not dedicated jailbreak benchmarks.

---

### Review · Reviewer_9FDk · 2026-03-13

**Summary Of Contributions:**

## Summary

This paper introduces InferenceGuard, an inference-time alignment method for LLMs that aims to generate safe responses with probability approaching one (almost surely) with respect to a given cost model. The key idea is to reformulate safe generation as a constrained MDP, augment the state space with a safety budget tracker to convert the constrained problem into an unconstrained one, and then shift optimization into the LLM's latent space for efficiency. The authors extend prior work on safety-augmented MDPs (Sootla et al., 2022) to the LLM setting, train a compact latent-space critic to guide beam search decoding, and demonstrate strong safety-reward tradeoffs across four models and three benchmarks.

## Strengths

- Novel and principled combination of safe RL state augmentation (Sootla et al., 2022) with inference-time LLM alignment — a setting that has received limited theoretical attention.
- Comprehensive experiments across four models with varying alignment levels, three benchmarks, and extensive ablations (Tables 5–7).
- Mostly sound theoretical framework with formal guarantees, extending prior results to the LLM setting.
- Well written

## Weaknesses

- The "almost sure safety" claim is heavily dependent on the cost model's quality, and this dependence is underemphasized throughout the paper.
- Key theoretical results (Theorems 1c, 2) rely on assumptions that are either unstated (injectivity of ϕ) or insufficiently discussed (existence of finite-cost policies).
- The non-negativity assumption on safety costs is potentially violated by practical cost models.
- Experiments are limited to 7–8B models with no comparison against safe RLHF fine-tuned baselines.
- BoN-Augmented, a much simpler method, performs surprisingly close to InferenceGuard in several settings.

**Audience:**

Yes

**Audience Explanation:**

With the rapid growth of LLMs and widespread use of GenAI in so many fields, safety is essential. I also feel that methods with rigorous theoretical guarantees are somewhat lacking in LLM literature so this is a refreshing change of pace.

**Broader Impact Concerns:**

There is already a broader impact statement which seems sufficient to me.

**Claims And Evidence:**

Yes

**Claims Explanation:**

Yes, but I have a few concerns that I'd like the authors to clarify.

## Major Comments

- **The "almost sure" safety guarantee is contingent on the cost model (Sections 1, 4, 7).** The theoretical guarantees (Theorem 2, p. 7) ensure constraint satisfaction within the MDP, but say nothing about whether the cost model captures real-world safety. If the cost model is miscalibrated or incomplete, the guarantees are vacuous. This is acknowledged in Limitations (p. 10) but relegating it to a brief end-of-paper paragraph feels insufficient given how prominently "almost surely safe" features in the title and abstract. The cost model is doing the vast majority of the heavy lifting, and this dependence should be discussed upfront in the introduction.

- **Experimental evaluation is limited to small models (Section 6, pp. 8–10).** All models tested are 7–8B parameters. I understand the compute constraints, but this raises a question: would InferenceGuard even be necessary on top of well-aligned frontier models that already achieve high safety rates? Since the method depends on the cost model's quality, it is not clear if we can achieve a highly accurate cost model which is still small. And if it needs to be big, then the frontier models probably have it baked in already. I think there is still room for low-cost/small models so a discussion of this would strengthen the paper.

- **Non-negativity assumption on safety costs is potentially restrictive (Section 4.2, p. 5).** The equivalence in Equation 5 relies critically on non-negative instantaneous costs. This may not hold in practice. Example: a model generates "This chemical compound is extremely dangerous" followed by "which is why it is strictly regulated and never sold to consumers" changes the safety profile entirely. If the cost model reflects these dynamics, incremental costs can be negative, and the theoretical guarantees break down. The authors should discuss when practical cost models satisfy this assumption.

- **Missing comparison with safe RLHF fine-tuned models (Section 6, pp. 8–10).** The paper positions itself as an alternative to RLHF (Section 1, p. 1) but never directly compares against a model fine-tuned with safe RLHF (e.g., Dai et al., 2023, which is cited). Without this comparison, it is difficult to assess whether inference-time alignment is a viable alternative or merely a complement to training-time methods.

- **Theorem 2's finite-cost assumption needs more discussion (Section 5.1, p. 7).** The theorem assumes "an optimal policy exists π* solving Equation 7 with a finite cost." This is a strong assumption. If every possible continuation of certain prompts inevitably violates the safety constraint under the cost model, the theorem is vacuous. What are sufficient conditions for finite-cost policies to exist? Is it sufficient that the base LLM assigns positive probability to at least one safe completion? Making this concrete would significantly strengthen the result.

- **Theorem 1(c): Doesn't this require injectivity of ϕ? (Section 5.1, p. 25, Eq. 40→41).** The proof claims that minimizing over latent policies π̄(ϕ(·)) is equivalent to minimizing over all original-space policies. But if ϕ is not injective i.e., two different token histories could map to the same latent state, then the latent policy cannot distinguish them, and the policy class {π̄(ϕ(·))} is strictly smaller than the full policy class. This would make the optimality claim hold only over a restricted set of policies. I think this is not an issue in practice, but it'll be good to clarify.

- **Claim about Lagrangian methods is too strong (Section 4.2, p. 4).** The paper states Lagrangian methods make it "challenging to maximize rewards while satisfying almost sure constraints optimally." This is not universally true — under appropriate convexity and other assumptions, Lagrangian methods can be optimal. The claim should be softened or made precise with respect to the specific setting here.

- **Baseline design choice for BoN and Beam Search (Section 6, p. 9).** Solutions are selected using c_task + λ · C_safety. A natural alternative: threshold C_safety (discard candidates exceeding the safety budget) and maximize c_task among the rest. This directly mirrors the constrained formulation, allows tracing the Pareto frontier by varying the threshold, and avoids the tuning problem the paper itself criticizes in Lagrangian approaches. Is there a reason this isn't compared?

- **BoN-Augmented is a strong baseline that deserves more discussion (Section 6, Figure 2).** BoN-Aug performs surprisingly well across multiple models — often close to InferenceGuard in the safety-reward plane, particularly on Beaver-7B and Vicuna-7B/HH-RLHF. Given that BoN-Aug is a much simpler method, the paper should engage more directly with when and why InferenceGuard's additional complexity (latent critic, diversity mechanism, retry logic) is justified over this simpler alternative.

- **Differentiation from Sootla et al. (2022) should be consolidated (Sections 2, 4.2, 5.1).** The related work is extensive and appreciated. However, since the state augmentation framework from Sootla et al. is the theoretical backbone, I would like a unified discussion explicitly delineating what is inherited versus what is new (LLM application, latent space, discrete vocabulary, Theorem 1). Currently this is scattered across multiple sections.

**Requested Changes:**

In addition to the concerns I raised above, here are a few minor comments about the writing.

## Minor Comments

- **Theorem 2's proof should be in the main text (Section 5.1, p. 7).** This is the paper's central result and the proof is short enough (~half a page) to include without significant space impact.

- **Notation inconsistencies:** C_safe vs. C_safety (compare Equation 3, p. 4 with surrounding text); z_{t=0} = 10 (p. 9) should be z_0 = 10.

- **Typos:**
  - "challengers" → "challenges" (p. 5)
  - "ht and ot ideal inputs" → "are ideal inputs" (p. 6)
  - "yt a generated token" → "is a generated token" (p. 3)
  - "Patero front" → "Pareto front" (pp. 2, 10)
  - "U* = V*" → "u = V*" (Appendix C.2, p. 24)

---

> ### Author Response · Authors · 2026-03-24
>
> We would like to thank the reviewer for the very detailed review and the helpful comments.
> > **The "almost sure" safety guarantee is contingent on the cost model.**
>
> We would like to clarify that both in abstract and intro, we always mention that our almost sure guarantees are w.r.t. given cost model (Abstract: "Consequently, we demonstrate formal safety guarantees w.r.t. the given cost model", Intro: guarantees the safety of
> LLM responses almost surely, i.e., with a probability approaching one w.r.t. a given cost model., we establish, for the first time, that for sufficiently large penalties one can guarantee almost sure safety in the original token space w.r.t. given cost model.).
>
> In order to clarify further, **we added the following sentence in the introduction**: Importantly, as in all constrained MDP formulations, our guarantees are defined with respect to the provided cost model, and thus depend on its ability to faithfully capture the desired notion of safety; our contribution is to ensure that such constraints, once specified, are satisfied almost surely at inference time.
>
> > **Experimental evaluation is limited to small models.**
>
> **We have added discussion and new larger-model results in the revised paper to address this point in Appendix E.2.**
>
> In principle, InferenceGuard is not tied to a particular model size or alignment level: its role is to provide an additional inference-time safety guarantee with respect to a given cost model. This is also reflected in our main experiments across different 7B LLaMA-based models. For example, Beaver-v3 is more safety-aligned than Alpaca, and InferenceGuard improves the safety rate of Beaver-v3 from 75.89\% to 100\%, while improving Alpaca from 29.47\% to 94.46\%; similarly, Vicuna is better aligned than Alpaca, and we observe gains on both. At the same time, we agree that applying the method to frontier models will likely require a stronger cost model, but a complete study on frontier models is beyond our current compute budget and time. We were able to conduct a partial validation on Llama-2-70B-Chat, using the same 7B cost model on HEx-PHI and PKU-SafeRLHF datasets. In this setting, the 70B base model starts with stronger reward and safety than the 7B--8B models, while InferenceGuard still achieves the highest safety rate among the compared baselines (97.33\% on HEx-PHI and 99.60\% on PKU-SafeRLHF). Overall, we view those results as preliminary but supportive evidence that the method can remain beneficial beyond small models, even when paired with a relatively small cost model.
>
> > **Non-negativity assumption on safety costs**
>
> In practice we only need costs to be bounded as one can always additively shift the cost values such that they are always non-negative. **We included this explanation in our limitations section.**
>
> > **Missing comparison with safe RLHF fine-tuned models**
>
> We do compare against the Safe-RLHF fine-tuned model in Section 6, namely Beaver-v3. Beaver-v3 is obtained by applying Safe-RLHF (Dai et al., 2023) to Alpaca-7B over three rounds of training on PKU-SafeRLHF. On the same PKU-SafeRLHF benchmark, InferenceGuard applied to Alpaca-7B achieves higher safety (88.14\% without critic and 94.46\% with critic) than the Safe-RLHF fine-tuned Beaver-v3 model (75.89\%). Moreover, InferenceGuard can also be applied on top of Beaver-v3 itself, further improving its safety rate, as shown in Figure 2 and Table 2. **We thank the reviewer for raising this point and have revised the evaluation section to make this comparison more explicit.**
>
> > **Theorem 2's finite-cost assumption needs more discussion**
>
> You are correct that the assumption requires the existence of a policy with exactly zero probability on all unsafe completions, i.e. a condition that standard LLMs cannot satisfy since they assign non-zero probability to essentially all sequences. However, the minimal assumptions for Theorem 2 to be meaningful are actually quite reasonable: (1) the base LLM assigns non-zero probability to at least one safe completion (a standard assumption satisfied by essentially all modern LLMs), and (2) we employ a test-time filter that assigns exactly zero probability to unsafe outputs. Under this interpretation, our policy space consists of filtered LLMs rather than bare autoregressive models, making the finite-cost assumption concrete.
>
> (to be continued...)

---

> ### Author Response · Authors · 2026-03-24
>
> > **Theorem 1(c): Doesn't this require injectivity of $\phi$?**
>
> We do not claim "minimizing over latent policies $\pi(\phi(·))$ is equivalent to minimizing over all original-space policies" and that is why we have carefully worded the proof as follows just below Eqn. 41 (page 26) : "the minimization of $\pi$ is over set of all policies covered by $\bar{\pi}(\phi())$. We agree that if $\phi$ is not injective this could potentially be smaller than the full policy class. However, we note that in practice, we define the latent state to be $h_t$ (hidden state) and $o_t$ (logits) of a particular token sequence/history. Since these representations are high-dimensional and depend on the full token history, they typically results in distinct latent states for different token sequences and are often injective as analyzed in [1].
>
> [1] Nikolaou, Giorgos, Tommaso Mencattini, Donato Crisostomi, Andrea Santilli, Yannis Panagakis, and Emanuele Rodolà. "Language models are injective and hence invertible." ICLR 2026.
>
> > **Claim about Lagrangian methods is too strong.**
>
> We agree that Lagrangian methods can, in principle, be used to address constrained optimization problems. **We have revised Section 4.2 to make clear that our concern is specific to the present setting rather than a general limitation of Lagrangian methods.** However, achieving an optimal solution to our constrained formulation in Eq. (3) with almost-sure guarantees presents several challenges in our setting: (i) Classical guarantees for Lagrangian methods rely on convexity and strong duality to ensure that optimizing the dual variable yields a constraint-satisfying solution. In our setting, where the policy is parameterized by an LLM, the problem is highly non-convex, and these guarantees do not apply. (ii) Our method operates at inference time, where the optimal Lagrange multiplier may depend on the specific prompt. Identifying such a multiplier would require iterative updates of $\lambda$, each involving a re-optimization over generated trajectories (e.g., via sampling or beam search), making the procedure significantly more expensive at inference time. (iii) Standard Lagrangian formulations enforce constraints in expectation and do not, in general, provide almost-sure (probability-one) guarantees. In contrast, our augmented-state formulation explicitly tracks the remaining safety budget and enables trajectory-level guarantees, which are more suitable for our setting.
>
> > **Baseline design choice for BoN and Beam Search.**
>
> The suggested alternative, which discard candidates that exceed the safety budget and then maximizing $c_{\text{task}}$ over the remaining canditates, is what exactly augmented Best-of-$N$ and augmented Beam Search baselines are designed to capture. These variants use a threshold on $c_{\text{safety}}$ and then select the candidate with the best task score among those satisfying the safety budget. We include both Lagrangian and augmented versions because Lagrangian relaxation is the most commonly used method in constrained and safe RL, while the augmented variants represent the more direct constrained alternative. This distinction is also one of the motivation of our method: instead of relying on the overly conservative soft constraint induced by Lagrangian tuning, we explicitly enforce the safety constraint and provide a corresponding theoretical guarantee. **We have revised the paper to make this baseline design clearer.**
>
> > **BoN-Augmented is a strong baseline**
>
> We want to highlight that both BoN-Augmented and Beam-Augmented are constrained variants introduced in this work, rather than standard baselines from prior decoding literature. They are simple and effective on safety-aligned models such as Beaver-v3 because safe candidates are likely to appear in the sampled pool in safety-aligned models. However, in riskier settings, especially for weakly aligned models such as Alpaca, where safe responses are not likely to emerge from sampling alone. There, additional intervention is required early in the decoding process. InferenceGuard uses lookahead search, explicit safety-budget tracking, and critic estimation of future risk to steer generation toward safe continuations before unsafe trajectories fully unfold. This is the main setting in which the added complexity of InferenceGuard is justified. **We have updated the main evaluation section to make this comparison and its implications clearer.**
>
> (to be continued...)

---

> ### Author Response · Authors · 2026-03-24
>
> > **Differentiation from Sootla et al. (2022)**
>
> **We updated the related prior to clarify the relationship to Sootla et al. (2022) in Appendix A.**
>
> While our work is related to prior state-augmentation approaches, it addresses a fundamentally different setting—namely, inference-time alignment for LLMs with a frozen model—rather than policy optimization in RL. Beyond this, our contributions differ in several key aspects: (i) we formulate safe LLM decoding as a constrained MDP over token generation, (ii) we introduce a latent-space augmented formulation and prove that guarantees transfer back to the original token space (Theorem 1), (iii) we adapt the analysis to the discrete action space (token vocabulary), (iv) we develop a practical inference-time decoding algorithm (InferenceGuard), (v) we introduce latent-space critic training for efficient inference-time guidance, and (vi) we provide almost-sure safety guarantees in the original token space.
>
> > **Theorem 2’s proof should be in the main text.**
>
> We moved the proof in the main body.
>
> > **Typos**
>
> Thank you, we fixed those.

---

> ### Comment · Reviewer_9FDk · 2026-04-06
> **Thank you for the thorough responses and edits!**
>
> I thank the authors for the detailed responses and for making the edits to their manuscript. I believe it makes it a stronger work and all my concerns have been addressed. I think it is a strong paper that would be a valuable contribution to TMLR.

---

### Review · Reviewer_zGaQ · 2026-04-27

**Summary Of Contributions:**

The paper introduces InferenceGuard, a novel approach for aligning Large Language Models (LLMs) at inference time to ensure they generate safe responses. Instead of fine-tuning the model's weights, the authors frame the generation of safe text as a constrained Markov Decision Process (cMDP) operating within the LLM's latent space.

The authors establish a mathematical framework that guarantees "almost sure" safety—meaning the probability of a safe response approaches one—with respect to a given safety cost model.  To bypass the limitations of traditional Lagrangian optimization, the authors map the constrained problem to an unconstrained one by adding a "safety state" variable that tracks the remaining safety budget dynamically.

Strength: Unlike traditional methods that only enforce safety constraints in expectation, this approach mathematically guarantees that safety constraints are satisfied at the individual trajectory level. By adjusting outputs purely at inference time, the method avoids the high computational costs and the risk of overfitting associated with training-time alignment techniques like RLHF. By operating in the latent space, the trained critic network remains small and fast, allowing InferenceGuard to match the decoding speed of standard beam search.


Weakness: The "almost sure" safety guarantees are strictly bound to the provided safety cost model. If the cost model is biased, inaccurate, or fails to recognize certain types of harm, the LLM's outputs will reflect those blind spots.  Because the method relies on multi-sample lookahead decoding (generating and evaluating multiple beams), it inherently introduces higher latency and computational overhead compared to simply generating a response from the base model. The rigorous mathematical proofs require the assumption that instantaneous safety costs are non-negative. While the authors note costs can be artificially shifted to satisfy this in practice, it is a strict condition for the theory to hold.

**Audience:**

Yes

**Audience Explanation:**

As the deployment of LLMs scales, ensuring they do not generate harmful content is a critical priority. The community is actively looking for alternatives to computationally expensive and overfitting-prone training methods like RLHF. This paper offers a compelling inference-time alternative that many applied researchers and engineers will want to study and potentially build upon.

**Broader Impact Concerns:**

There is no concerns on the ethical implications of the work.

**Claims And Evidence:**

Yes

**Claims Explanation:**

1. The authors do not merely propose a heuristic approach; they provide rigorous mathematical backing for their claims. They establish that tracking a safety state enables the mapping of a constrained MDP into an unconstrained one within the LLM's latent space. The paper provides formal proofs (Theorem 1 and Theorem 2) demonstrating that optimizing this augmented MDP with sufficiently large penalties guarantees "almost sure" constraint satisfaction in the original token space. This substantiates the titular claim of "almost surely safe" generation.
2. The empirical evidence clearly supports the claim that InferenceGuard achieves a state-of-the-art balance between task reward and safety. The results demonstrate exceptionally high safety rates (e.g., 98.45% on Llama3.1-8B-Instruct and 100% on Beaver-v3-7B) while maintaining high task rewards. Scatter plots and Pareto curves effectively visualize how InferenceGuard dominates the Pareto front compared to the baselines, avoiding the severe reward drops seen in Lagrangian-based methods.

**Requested Changes:**

There is no major problems. I suggest move the discussion of the "alignment tax" and inference latency from the Appendix to the main text.

---

> ### Author Response · Authors · 2026-04-30
>
> We would like to thank the reviewer for the thoughtful and positive evaluation of our work, and for recognizing the proposed InferenceGuard framework as well-motivated, mathematically grounded, and empirically supported. We also thank the reviewer for the helpful suggestion regarding the discussion of the “alignment tax” and inference latency. These aspects are important for understanding the practical implications of InferenceGuard, and we have therefore moved the latency analysis together with the discussion of the “alignment tax” from the appendix to the main text (see page 11).

---

### Decision · Action_Editor_cYBj · 2026-05-12

**Recommendation:** Accept as is

**Audience:**

Yes

**Audience Explanation:**

Inference-time safety alignment with formal guarantees is a timely and practically relevant topic for both the safe RL and LLM deployment communities, and the constrained-MDP perspective offers a principled framework that complements existing training-time approaches.

**Claims And Evidence:**

Yes

**Claims Explanation:**

The theoretical guarantees are rigorously established under clearly stated assumptions, and the empirical evaluation across multiple models, benchmarks, and ablations consistently supports the claimed safety–reward trade-offs, with three of four reviewers affirming that claims are well-supported.